# A Retrospective Study in Colorectal Adenocarcinoma Uncovers the Potential of Circ-CCT3 as a Predictor of Tumor Recurrence

**DOI:** 10.3390/biomedicines13102432

**Published:** 2025-10-06

**Authors:** Panagiotis Kokoropoulos, Spyridon Christodoulou, Panagiotis Tsiakanikas, Efthimios Poulios, Panteleimon Vassiliu, Christos K. Kontos, Nikolaos Arkadopoulos

**Affiliations:** 1Fourth Department of Surgery, University General Hospital “Attikon”, National and Kapodistrian University of Athens, 12462 Athens, Greece; kokoropoulos@yahoo.gr (P.K.); spyridon.christodoulou@yahoo.gr (S.C.); efthimios.poulios@gmail.com (E.P.); pant.greek@gmail.com (P.V.); narkado@hotmail.com (N.A.); 2Department of Biochemistry and Molecular Biology, Faculty of Biology, National and Kapodistrian University of Athens, Panepistimiopolis, 15771 Athens, Greece; ptsiak@biol.uoa.gr

**Keywords:** colon cancer, circular RNA (circRNA), molecular tumor markers, prognosis, prognostic biomarkers, non-coding RNA

## Abstract

**Background/Objectives:** Colorectal cancer (CRC) is one of the most prevalent malignancies; this issue underlines the need for accurate molecular biomarkers for early detection and accurate prognosis. Circular RNAs (circRNAs) have recently emerged as very promising cancer biomarkers. The circular transcript of the chaperonin-containing TCP1 subunit 3 (*CCT3*) gene, namely circ-CCT3, is a significant oncogenic driver. In gastrointestinal malignancies, circ-CCT3 promotes tumor growth by sponging tumor-suppressor miRNAs. In this study, we examined whether circ-CCT3 expression can predict the prognosis of patients diagnosed with colorectal adenocarcinoma, the most frequent type of CRC. **Methods:** Total RNA was extracted from pulverized, fresh frozen colorectal tissues and reverse-transcribed. A previously developed, highly sensitive quantitative PCR (qPCR) assay was applied to determine circ-CCT3 expression in 216 primary colorectal adenocarcinoma tissue specimens and 86 paired normal colorectal tissues. **Results:** circ-CCT3 was significantly upregulated in colorectal adenocarcinoma tissues, in comparison to their non-cancerous tissue counterparts. Higher circ-CCT3 expression was associated with a poorer disease-free (DFS) and overall survival (OS) of colorectal adenocarcinoma patients. Interestingly, multivariate Cox regression showed that the prognostic value of circ-CCT3 expression regarding DFS was independent of other established prognosticators used in clinical practice, including TNM staging. Furthermore, the stratification of patients based on the TNM classification of the tumors revealed that increased circ-CCT3 levels predicted shorter DFS and OS intervals, especially in the subgroup of TNM stage II or III patients. **Conclusions:** Our study provides evidence that circ-CCT3 overexpression constitutes a promising molecular biomarker of poor prognosis in colorectal adenocarcinoma, independently predicting tumor recurrence.

## 1. Introduction

Approximately 1.9–2.0 million new cases of colorectal cancer (CRC) are diagnosed globally each year, making it worldwide the third most common human malignancy. This high incidence leads to an estimated 900,000 annual deaths, placing CRC as the second leading cause of cancer-related mortality and highlighting its significant lethality [1,2,3]. Over 90% of CRCs are adenocarcinomas developing from benign adenomatous polyps in the colonic mucosa [4]. The primary and potential curative treatment of colon cancer is surgical resection, followed by adjuvant chemo- and/or radiotherapy [5,6]. About 20% of patients already have metastatic disease at their initial presentation, and up to half of those diagnosed with localized tumors will experience distant relapse [7]. This highlights the urgent need for novel molecular biomarkers to improve screening, risk stratification, and treatment selection. TNM staging remains the standard prognostic tool guiding the clinical management of colorectal cancer patients. Recently, liquid biopsy and stool DNA assays have facilitated the non-invasive detection of minimal residual disease, recurrence risk estimation, and therapy response. Additionally, molecular biomarkers like *APC*/*RAS*/*BRAF* mutations, microsatellite instability (MSI) status, and CpG island methylator phenotype (CIMP) have enabled personalized management, addressing some limitations of conventional approaches [8,9].

Human chaperonin-containing TCP1 subunit 3 (CCT3) is an essential component of the eukaryotic chaperonin T-complex protein ring complex (TRiC), which consists of two stacked, asymmetric rings forming a barrel-shaped structure. TRiC encapsulates unfolded polypeptides and facilitates their ATP-dependent folding. As an ATP-dependent molecular chaperone, TRiC mediates the folding of about 10% of cytosolic proteins [10]. CCT3 deregulation disrupts proteostasis, contributing to cancer development and progression [11]. In CRC cell models, CCT3 depletion reduces cellular viability and impairs colony formation, hindering tumor growth [12,13]. While *CCT3* exhibits moderate basal expression in gastrointestinal epithelium, it is markedly overexpressed in CRC, with both mRNA and protein levels found significantly elevated compared to adjacent non-tumor tissues [12]. Clinically, elevated CCT3 protein and/or mRNA expression can discriminate malignant from benign tissue specimens and be associated with an advanced disease stage and the poor survival of patients [12,14].

Alternative splicing is a precisely regulated mechanism, that generates diverse mRNA transcript variants and protein isoforms through combinatorial exon selection in multi-exon genes. Back-splicing covalently links a downstream 5′ splice donor site to an upstream 3′ splice acceptor site, generating circular RNAs (circRNAs) lacking typical 5′ and 3′ ends [15]. While many circRNAs remain functionally uncharacterized, emerging evidence unveils their multifaceted roles in post-transcriptional regulation, including the modulation of parental pre-mRNA maturation, interactions with transcriptional machinery, microRNA (miRNA) sequestration, and RNA-binding protein (RBP) decoy activity [16]. Of note, their circular conformation confers intrinsic ribonuclease resistance; the prolonged half-life of circRNAs in bodily fluids, as well as their differential expression in cancer patients renders these molecules promising clinical biomarkers for non-invasive diagnosis, survival estimation, and treatment-response monitoring. In CRC, aberrantly expressed circRNAs drive critical oncogenic features, such as uncontrolled proliferation, apoptosis resistance, and metastatic progression [16].

CCT3 is a subunit of the chaperonin TRiC complex, essential for proteostasis and the folding of cytosolic proteins. The dysregulation of CCT3 disrupts proteostasis, promoting cancer development [17]. Notably, circ-CCT3, which originates from the back-splicing of the *CCT3* pre-mRNA, has been implicated in the pathogenesis and progression of several human malignancies [18,19,20,21,22,23,24]. circ-CCT3 functions as a competitive endogenous RNA (ceRNA), sponging tumor-suppressive miRNAs and subsequently modulating the expression of oncogenic transcripts. In pancreatic and colorectal cancer, circ-CCT3 sequesters miR-613, which typically represses WNT3 and VEGFA, enhancing Wnt/β-catenin signaling and promoting angiogenesis. In hepatocellular carcinoma (HCC), circ-CCT3 sponges miR-378a-3p restoring fms-related receptor tyrosine kinase 1 (FLT1) expression and accelerating HCC progression [19]. Similarly, circ-CCT3 represses miR-1287-5p to drive TEA domain transcription factor 1 (TEAD1) overexpression and the subsequent activation of patched 1 (PTCH1) and lysyl oxidase (LOX) and to promote liver tumorigenesis [20]. In multiple myeloma, circ-CCT3 binds miR-223-3p, attenuating the miR-223-3p-mediated repression of bromodomain containing 4 (BRD4) to sensitize cells to bortezomib treatment [18].

Emerging evidence designates circ-CCT3 as a significant oncogenic driver across multiple cancers [18,19,20,21,22,23,24]. For example, in several gastrointestinal malignancies, circ-CCT3 promotes tumor growth by sponging tumor-suppressor miRNAs such as miR-1287-5p, miR-378a-3p and miR-613 [19,20,22]. In CRC, circ-CCT3 is overexpressed and sponges miR-613, directly upregulating miR-613 downstream targets Wnt family member 3 (WNT3) and vascular endothelial growth factor A (VEGFA). This upregulation correlates with a larger tumor size and microvascular invasion, indicating a critical biological role in the malignant transformation and progression of CRC [24]. Beyond its mechanistic roles, circ-CCT3 shows clinical promise as both a diagnostic and prognostic biomarker. In bladder cancer, it can effectively distinguish malignant from healthy tissue and correlates with adverse outcomes [25]. Moreover, its expression also exhibits prognostic significance in multiple myeloma, enabling the better risk stratification of stage II myeloma patients [23], underscoring the translational value of circ-CCT3 for precision oncology in solid and hematological malignancies.

Based on the established multipurpose role of circ-CCT3 as an oncogenic driver and clinical biomarker, this study investigates for the first time its diagnostic and prognostic utility in CRC. For this purpose, we developed a reverse-transcription quantitative PCR (RT-qPCR) assay for the relative quantification of circ-CCT3 expression in CRC tissues and matched adjacent non-cancerous specimens.

## 2. Materials and Methods

### 2.1. Sample Collection

The biobank of tissue samples used in this retrospective study consisted of 216 primary colorectal adenocarcinoma specimens from patients who previously underwent surgery at the University General Hospital “Attikon”. Exclusion criteria included receiving neoadjuvant therapy prior to surgery, missing clinicopathological data, and the existence of known comorbidities at the time of surgery. Inclusion criteria included the diagnosis of primary colorectal adenocarcinoma and sufficient mass of cancerous colorectal tissue available (i.e., 10–50 mg of tissue). To randomly select patients from the larger cohort of those that could be enrolled in this study, we applied the common method of simple random sampling. Thus, after having defined the population of interest and determined the sample size, we assigned unique numbers prior to the random selection of numbers produced by a random number generator. Lastly, we identified the selected specimens. Their number represents approximately 30% of the newly diagnosed patients with primary colorectal adenocarcinoma at the University General Hospital “Attikon” during the respective period of tissue specimen collection. At this point, it should be mentioned that adjacent non-cancerous colorectal tissue specimens were also available only for 86 out of the 216 cases.

After the surgical resection of each colorectal tumor, each cancerous tissue specimen was histologically evaluated by a pathologist and immediately frozen in liquid nitrogen. For 86 cases, adjacent non-tumorous colorectal tissue was also collected. This study received the approval of the institutional ethics committee of the University General Hospital “Attikon” (approval number: 31; date 29 January 2009) and was carried out in accordance with the Helsinki Declaration. All patients were informed in detail about the purpose of the research and consented to provide tissue sample(s).

The clinicopathological data comprised the size and location of the colorectal tumors, the histological grade, and the TNM stage. TNM staging is based on three parameters, including tumor invasion (T), regional lymph node status (N), and the presence or absence of distant metastases (M). The collected follow-up information comprised the cause of death for those whose death was due to colorectal adenocarcinoma, the recurrence of the disease, and the respective dates.

### 2.2. Total RNA Extraction and Reverse Transcription

All fresh frozen colorectal tissue specimens were pulverized. Total RNA was isolated using TRIzol^®^ Reagent (Ambion™, Thermo Fisher Scientific Inc., Waltham, MA, USA) and immediately stored at −80 °C, for further use. The determination of the concentration and quality control of the total RNA extracts were performed with a BioSpec-nano Micro-volume UV–Vis Spectrophotometer (Shimadju, Kyoto, Japan). RNA integrity was determined by agarose gel electrophoresis.

Next, first-strand cDNA was produced by M-MLV reverse transcriptase (Invitrogen™, Thermo Fisher Scientific Inc., Carlsbad, CA, USA) and random hexamers (New England Biolabs Ltd., Hitchin, UK) as primers, according to the manufacturer’s instructions. The mass of each RNA sample used in the reverse transcription was 2 µg. First-strand cDNA synthesis was performed in a MiniAmp Thermal Cycler (Applied Biosystems™, Thermo Fisher Scientific Inc.). Total RNA extracted from a randomly selected CRC cell line, Caco-2, was employed as positive control along with each batch of total RNA extracts of tissue specimens; a negative control was also included in each run.

### 2.3. Relative Circ-CCT3 Quantification, Using Real-Time Quantitative Polymerase Chain Reaction (qPCR)

Due to the low expression levels of circ-CCT3, a first-round PCR assay was conducted for the pre-amplification of these molecules and of *GAPDH* mRNA, which was used as an internal control gene. A real-time qPCR assay was then applied for the relative quantification of circ-CCT3, based on a previously described protocol developed by the members of our research group. The previous real-time qPCR assay optimization included designing specific divergent primers for the circ-CCT3 pre-amplification and qPCR, as well as convergent primers for *GAPDH* amplification, the optimization of the conditions and standardization of the molecular quantification assays, and the Sanger sequencing of the amplicon spanning circ-CCT3 back-splice junction [23]. In our study, each qPCR reaction was performed in duplicates, to assure the reproducibility of the obtained data. All primers used are presented in Table 1.

The expression levels of circ-CCT3 were determined by applying the comparative Ct (2^−∆∆Ct^) method [26]. For this purpose, *GAPDH* was used as an internal reference gene to normalize the PCRs for the quantity of RNA added to the cDNA synthesis reactions; Caco-2 cDNA was used as a calibrator for the rendering results from distinct qPCR runs comparable. Moreover, the pre-amplified Caco-2 cDNA accompanying each batch of samples was used for minor corrections of Ct variability resulting from the pre-amplification step. Thus, the inclusion of appropriate controls in each reaction (i.e., reverse transcription, pre-amplification, and qPCR) excluded any potential experimental bias. The relative expression of circ-CCT3 circRNA in each sample was determined in RQUs by calculating the ratio of its levels to *GAPDH* transcripts, divided by the same ratio calculated for the calibrator sample. Additionally, melting curves were generated to confirm circRNA-specific amplification in each sample.

### 2.4. Biostatistical Analysis

Extensive biostatistical analysis was conducted with the IBM SPSS Statistics software (version 29) (IBM Corp., Armonk, NY, USA). The variables of clinicopathological data analyzed in the current study included patients’ tumor location, histological grade, TNM staging and its components, as well as treatment with radiotherapy and/or chemotherapy after surgery. Moreover, our study included patients’ survival data, as described later in this section.

The distributions of circ-CCT3 levels in both non-cancerous and cancerous colorectal tissue samples were non-Gaussian; therefore, only non-parametric tests (the Mann–Whitney *U* test and Jonckheere–Terpstra test, where appropriate) were applied to assess the significance of differences in circ-CCT3 expression observed among patients’ subgroups, stratified according to their tumor location and TNM stage. Moreover, the Wilcoxon signed-rank test was applied to assess the significance of differences in circ-CCT3 expression between the 86 pairs of cancerous and adjacent non-cancerous colorectal tissues. Receiver operating characteristic (ROC) curve analysis was also performed to evaluate the ability of circ-CCT3 expression to distinguish colorectal adenocarcinoma from normal colorectal tissue. The ROC curve analysis was based on the cohort of 216 cancerous tissue specimens and the cohort of 86 non-cancerous tissue specimens.

Next, the expression levels of circ-CCT3 were split at the median, and patients were thus categorized into two groups (high vs. low expressors). Tumor recurrence was considered as the endpoint for disease-free survival (DFS) analysis, while death due to CRC was considered as the endpoint for overall survival (DFS) analysis. Survival data were unavailable for 18 of the 216 patients, because these 18 patients were lost from follow-up very soon after surgery; these patients were hence excluded from the survival analysis only. Twenty-six (26) patients out of the remaining 198 with complete follow-up data had distant metastasis (M1) at the time of surgery and were hence additionally excluded only from the DFS analysis, as they are not considered, in general, to achieve complete remission after tumor resection. Therefore, the DFS analysis included 172 out of the 216 patients; 61 relapses were recorded during the accrual follow-up period and considered as events for DFS. Moreover, 198 out of the 216 patients were included in the OS analysis; 91 deaths related to CRC were recorded during the accrual follow-up period and considered as events for OS.

Kaplan–Meier survival analysis was performed to examine any association between circ-CCT3 expression and patient outcomes with regard to DFS and OS. Besides that, we stratified our patients according to their tumor location (colon or rectum) and again performed Kaplan–Meier survival analysis, separately in each patients’ subgroup. The Mantel–Cox (log-rank) test was used to calculate *p* values for any differences observed from the comparison of Kaplan–Meier curves. To further assess the prognostic potential of circ-CCT3 expression and estimate the hazard ratio (HR) for disease recurrence and disease-related death, the univariate Cox regression analysis for each variable was carried out; 95% confidence intervals (CIs) were calculated for all estimated HRs, along with the respective *p* values. Subsequently, multivariate Cox regression models adjusted for all significant clinicopathological prognosticators were built. Cox regression analyses were performed using 1000 bootstrap samples. The bootstrap bias-corrected and accelerated (BCa) method was chosen to calculate the bootstrap *p* values and 95% CIs for each estimated HR.

The statistical significance level in each statistical test was set as lower than 0.050 (*p* < 0.050).

## 3. Results

### 3.1. Circ-CCT3 Expression Is Overexpressed in Colorectal Adenocarcinoma, Compared to Adjacent Non-Cancerous Tissues

The median age of colorectal adenocarcinoma patients was 68 years old (range: 35–93; interquartile range: 58–75). The clinicοpathological characteristics of the patients’ tumors are presented in Table 2. The majority of patients had a tumor in the colon. Among the 216 colorectal adenocarcinoma patients, over half of them had received chemotherapy after surgery. Most adenocarcinomas were characterized as histological grade II and TNM stage III tumors.

circ-CCT3 levels in colorectal adenocarcinoma samples ranged from 2.07 to 7.97 RQU with a mean ± SEM of 5.13 ± 0.123, whereas in non-cancerous tissues, the levels fluctuated from 0.814 to 3.76 RQU with a mean ± SEM of 2.25 ± 0.099 (Table 3). Comparison of circ-CCT3 expression levels between pairs of cancerous and normal tissues showed profound upregulation (*p* < 0.001) in the former (75 out of 86 tissue pairs; 87.2%) (Figure 1).

Moreover, the receiver operating characteristic (ROC) curve analysis demonstrated that the circ-CCT3 expression may very efficiently distinguish colorectal adenocarcinoma from normal colorectal mucosa [area under the ROC curve (AUC) = 0.92, 95% CI = 0.89–0.95, *p* < 0.001] (Figure 2).

circ-CCT3 expression differed also significantly among subgroups of patients of a different TNM stage (*p* < 0.001). More specifically, the expression of this circRNA was upregulated in malignant tumors of patients with nodal metastases (N1 and N2 vs. N0) and advanced TNM stage (Figure 3A and Figure 3B, respectively).

After the categorization of patients into two categories (high vs. low expressors) by splitting circ-CCT3 expression at the median value, the expression status of this circRNA was shown to be significantly associated with nodal status (*p* = 0.004) and TNM stage (*p* = 0.006). On the other hand, no significant association was observed between circ-CCT3 expression status and tumor location, histological grade, tumor invasion (T), or the presence of distant metastasis (M).

### 3.2. Circ-CCT3 Overexpression Predicts Poor DFS and OS in Colorectal Adenocarcinoma

Next, we investigated whether there was any correlation between circ-CCT3 expression and the DFS and/or the OS of colorectal adenocarcinoma patients. In the DFS analysis, 172 colorectal adenocarcinoma patients with no distant metastasis were included. Among these, 61 patients (35.5%) had tumor recurrence during the accrual follow-up period. The OS analysis comprised 198 colorectal adenocarcinoma patients with complete follow-up data, out of whom 91 patients (45.6%) succumbed to their disease during the accrual follow-up period. The median follow-up time was 98 months.

Colorectal adenocarcinoma patients with circ-CCT3 overexpression were shown to have significantly lower DFS probabilities (*p* < 0.001), compared to patients with lower circ-CCT3 levels (Figure 4A). Additionally, the unfavorable prognostic value of circ-CCT3 overexpression was observed in Kaplan–Meier OS analysis, which revealed that colorectal adenocarcinoma patients with circ-CCT3 overexpression had significantly poorer OS (*p* < 0.001) (Figure 4B). These results were also confirmed by univariate Cox regression analysis, estimating a hazard ratio (HR) of 2.35 (*p* = 0.001) for colorectal adenocarcinoma patients’ relapse with tumors overexpressing circ-CCT3, and an HR of 2.12 (*p* < 0.001) for cancer-related death, in contrast to those with lower levels of circ-CCT3 (Table 4).

After the stratification of colorectal adenocarcinoma patients according to their tumor location, we noticed that patients with colon tumors (122 for DFS and 136 for OS) and the overexpression of circ-CCT3 showed significantly shorter DFS intervals (*p* = 0.032) and OS intervals (*p* = 0.005) (Figure 5A,B). Similarly, as shown in Figure 5C,D, colorectal adenocarcinoma patients with rectal tumors (50 for DFS and 62 for OS) and lower levels of circ-CCT3 had significantly higher survival probabilities with regard to both DFS (*p* = 0.014) and OS (*p* = 0.038).

### 3.3. Circ-CCT3 Overexpression Predicts Short-Term Relapse in Colorectal Adenocarcinoma, Independently of Other Established Prognosticators

Multivariate Cox regression models comprised the tumor location, histological grade, TNM stage, and the type of treatment received after tumor resection (radiotherapy or chemotherapy), as presented in Table 4. circ-CCT3 overexpression was shown to retain its unfavorable prognostic significance regarding DFS, independently of other established prognostic factors in colorectal adenocarcinoma (HR = 1.75; *p* = 0.039). However, its prognostic power concerning OS did not prove to be independent (*p* = 0.33) of other important prognosticators (Table 5).

## 4. Discussion

Colorectal cancer (CRC) remains a significant global health challenge, being the third most prevalent cancer and the second leading cause of cancer-related deaths worldwide [27]. Even with advances in screening programs and treatment strategies, there are still considerable clinical shortcomings. Approximately 20% of patients present with metastatic disease at initial diagnosis, and nearly half of those whose tumors are originally confined to the colon later develop distant recurrence [7]. Such findings reveal that current staging systems possess limited clinical efficacy for precise risk evaluation and tailored prognostic assessment. Despite the implementation of advanced molecular classification systems, such as the consensus molecular subtypes (CMS), which bridge conventional staging with molecular features, these frameworks still inadequately capture intratumoral heterogeneity or accurately predict therapeutic resistance [8,28]. This is more evident in local and locally advanced CRC (stages II–III), where the clinical outcomes vary despite comparable clinicopathological characteristics [8,29,30]. Therefore, it is critical to discover novel biomarkers that accurately reflect the molecular background of the disease and improve both diagnostic precision and prognostic reliability.

circRNAs have recently been suggested as candidate molecular biomarkers for CRC diagnosis and prognosis, due to their increased stability and half-life [16]. For example, increased CDR1as expression has been correlated with tumor size, TNM stage, and the poor overall survival of CRC patients, rendering it a potential prognostic biomarker [31]. Another study highlighted a group of three circulating circRNAs in plasma with diagnostic value, independent of carcinoembryonic antigen (CEA) and CA19-9. Thus, the circulating circ-CCDC66, circ-ABCC1, and circ-STIL levels are significantly lower in the plasma of CRC patients, compared to healthy controls. circ-CCDC66 and circ-ABCC1 levels were also decreased in precursor lesions of CRC and were able to diagnose early-stage CRC. Furthermore, low circ-ABCC1 expression was associated with tumor growth and disease progression [32], while the overexpression of circ-PRMT1 in colorectal adenocarcinoma has very recently been shown to predict recurrence and poor OS [33], similarly to circ-CCT3.

In the current study, we developed an qPCR assay using divergent primers designed to specifically span the back-splice junction of circ-CCT3. This design is critical for the selective quantification of circ-CCT3, as it excludes the amplification of linear *CCT3* transcripts that share common exonic sequences. Furthermore, a preamplification step was included to address the typically low initial abundance of circ-CCT3. This strategy enabled the accurate quantification of circ-CCT3 and subsequent clinical evaluation of circ-CCT3 expression as a potential biomarker in CRC. Our results indicated a profound circ-CCT3 upregulation in CRC tissues compared to matched non-cancerous colorectal mucosa in 87.2% of paired samples. This observation is in line with prior reports of elevated circ-CCT3 expression in other gastrointestinal malignancies [19,20,22]. More importantly, the ROC analysis revealed exceptional discriminatory power, surpassing the performance of conventional protein biomarkers like CEA and stool DNA tests [34]. This high AUC underscores the potential of circ-CCT3 as a tissue-based diagnostic tool. Its intrinsic stability further suggests promise for non-invasive detection in liquid biopsies and fecal samples, which could revolutionize early screening strategies. It has recently been suggested that emerging biomarkers such as circRNAs and long non-coding RNAs (lncRNAs) offer opportunities for early CRC detection [35]. Beyond its potential diagnostic utility, higher circ-CCT3 expression levels were associated with a higher TNM stage and node involvement of CRC patients, indicating that circ-CCT3 levels increase progressively with disease severity. This finding is consistent with the modulation of VEGFA and Wnt/β-catenin signaling by circ-CCT3 to enhance CRC metastasis through miRNA sponging, which has recently been documented [24].

Based on findings presented in the current study, it would be tempting to detect and quantify circ-CCT3 levels in extracellular vesicles of CRC patients. Taking it a step further, it would be very interesting to study circ-CCT3 function and role in CRC. In fact, only a few circRNAs found in extracellular vesicles and produced specifically by cancerous or other cancer-associated cells, such as CRC-associated fibroblasts have been studied so far. Interestingly, such circRNAs have very important regulatory roles, also affecting post-transcriptional modifications of mRNAs [36]. Beyond its expression levels, the study of N^6^-methyladenosine (m^6^A) modifications of circ-CCT3 could prove very interesting, as this dynamic modification system participates in virtually all aspects of circRNA metabolism, including biogenesis, intracellular trafficking, and turnover [37]. By integrating mechanistic insights with preclinical results regarding circ-CCT3 expression in CRC, clinical translation barriers could be addressed.

The direct association of the increased circ-CCT3 levels with the dismal prognosis of patients has been previously reported in the case of other gastrointestinal malignancies, further confirming the observed results in CRC [19,20,22]. In this study, we report that circ-CCT3 overexpression emerged as a powerful independent predictor of poor DFS, conferring a 3-fold increased relapse risk of CRC patients. The Kaplan–Meier analysis of patients with localized disease further confirmed that this notion revealing a significantly shorter DFS in those characterized by elevated circ-CCT3 expression. This adverse prognostic impact was consistently retained within subgroups of patients stratified for important clinicopathological factors. Notably, increased circ-CCT3 expression levels in TNM stage II and III tumors were associated with a significantly higher relapse risk, indicating a potential utility in refining therapeutic decisions for these heterogeneous patient cohorts. Consistent with DFS findings, Kaplan–Meier analysis confirmed a significantly worse OS in CRC patients with high circ-CCT3 expression levels. This trend persisted regardless of primary tumor location and remained a robust predictor of adverse outcomes for patients with stage II/III tumors. Therefore, the quantification of circ-CCT3 expression levels could act as a surrogate biomarker in addition to TNM staging, for the determination of a subgroup of patients with a higher probability of disease recurrence. While the unfavorable prognostic value of the circ-CCT3 status is independent of important prognosticators with regard to DFS, this is not the case for the prediction of OS. The fact that the unfavorable prognostic value of circ-CCT3 status did not prove to be independent in the multivariate Cox regression analysis could be attributed to its significant association with TNM staging, which is a very strong prognosticator of OS. The independent prediction of short-term relapse but not of disease-related death has also been shown for other candidate biomarkers in colorectal adenocarcinoma, including kallikrein-related peptidase 4 (KLK4) mRNA and miR-24-3p expression [38,39]. In fact, several post-recurrence factors, such as the site and timing of recurrence, tumor characteristics like mucinous histology, and treatment options like surgery and chemotherapy, can significantly influence post-recurrence survival in colorectal cancer. For instance, the location of recurrence and the presence of lymph node metastasis can negatively affect OS of CRC patients. Moreover, the accurate and early detection of recurrent lesions is crucial for initiating appropriate treatment and can improve post-recurrence survival [40,41,42,43,44].

While demonstrating the potential clinical utility of circ-CCT3 in CRC, this study has inherent constraints largely derived from patient cohort composition. This is a retrospective, single-center study, based on colorectal tissue samples. Prospective multi-center validation in larger cohorts is a prerequisite to confirm clinical utility. Another limitation of this research study is the lack of information regarding *APC*, *KRAS*, and *BRAF* mutational status, MSI status, and CIMP, which did not allow categorization according to CMS widely used in clinical practice today [45]. The lack of information about tumor budding, a well-established independent prognostic factor in CRC [46], is another limitation of our study. This is due to the fact that most patients providing samples for this study were diagnosed and had their colorectal tumor removed before the establishment of this strong prognosticator. Finally, detailed information regarding neo-adjuvant therapy offered to some patients and comorbidities at the time of diagnosis are two parameters that should be taken into account in a future study. Nevertheless, it should be considered that cancer represents an ecological process as well. Addressing clinical challenges such as recurrence and treatment resistance may require perspectives that extend beyond genetic and molecular mechanisms alone. For instance, it is common knowledge that polymorphic microbiomes play important roles in CRC occurrence and development [47]. Additionally, information on the association between socioeconomic status and cancer is useful for policy-based CRC control [48].

## 5. Conclusions

Our study provides robust evidence for the clinical potential of circ-CCT3 expression as a prognostic biomarker in CRC. Elevated circ-CCT3 expression levels independently predict poor DFS/OS and discriminate cancerous from non-cancerous tissues. This sets the basis for prospective multi-center validation in larger patient cohorts to confirm clinical utility. Expanding analyses to liquid biopsies and fecal samples will elucidate non-invasive applicability, while functional studies should define its role in CRC heterogeneity and therapeutic resistance.

## Figures and Tables

**Figure 1 biomedicines-13-02432-f001:**
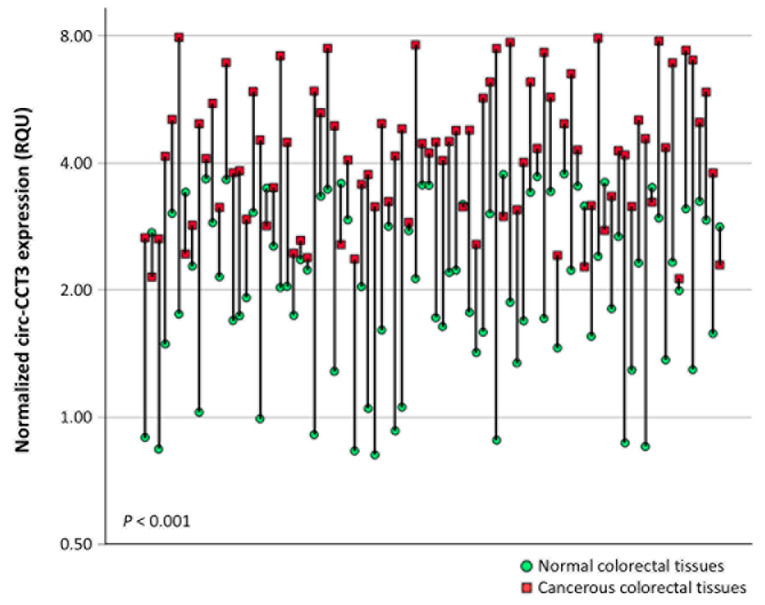
Comparison of the circ-CCT3 expression levels in 86 cancerous vs. normal adjacent colorectal tissue samples. The *p* value was estimated by the Wilcoxon signed-rank test.

**Figure 2 biomedicines-13-02432-f002:**
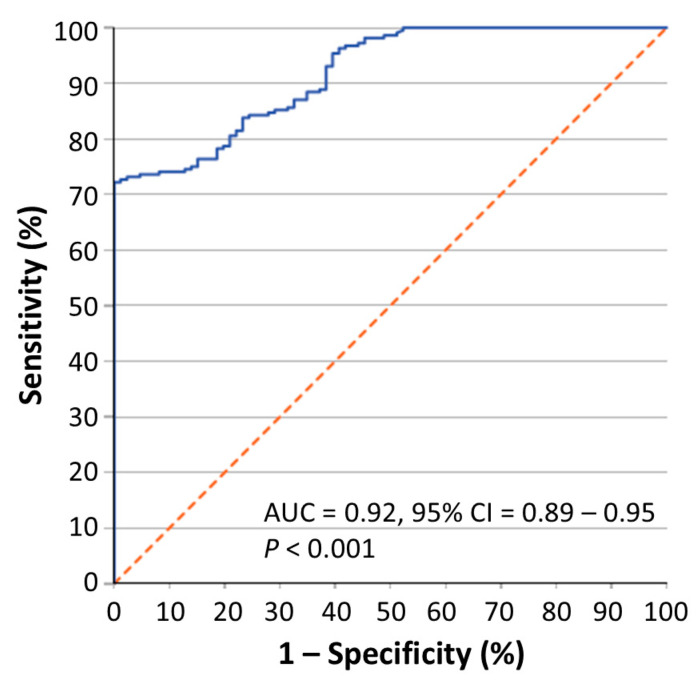
Receiver operating characteristic (ROC) analysis curve depicting the ability of circ-CCT3 expression to efficiently distinguish colorectal adenocarcinoma from adjacent non-cancerous colorectal tissue. Abbreviations: AUC, area under the ROC curve; CI, confidence interval. The *p* value was estimated by the Mann–Whitney *U* test.

**Figure 3 biomedicines-13-02432-f003:**
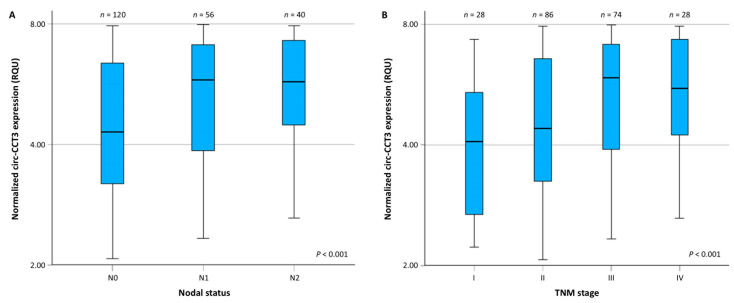
Comparison of the distribution of circ-CCT3 expression levels between colorectal adenocarcinomas from node-negative (N0) and node-positive (N1 or N2) patients (**A**), as well as among patients of different TNM stages (**B**), showed that circ-CCT3 overexpression was associated with positive regional lymph nodes and advanced TNM stages. The *p* value was estimated using the Jonckheere–Terpstra test. The line bars represent the median value (50th percentile) for each cohort, the bottom and top lines of each box indicate the 25th and 75th percentiles, respectively, and the whiskers extend to 1.5 times the height of each box.

**Figure 4 biomedicines-13-02432-f004:**
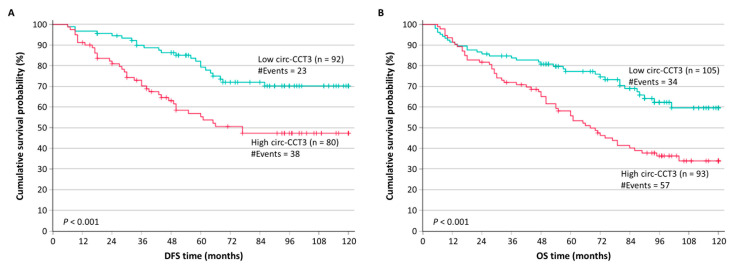
Kaplan–Meier survival curves for the disease-free survival (DFS) and overall survival (OS) of colorectal adenocarcinoma patients. Patients with colorectal adenocarcinomas overexpressing circ-CCT3 had significantly poorer DFS (**A**) and OS (**B**) than patients with neoplasms expressing low circ-CCT3 levels. The *p* values were estimated by the Mantel–Cox (log-rank) test.

**Figure 5 biomedicines-13-02432-f005:**
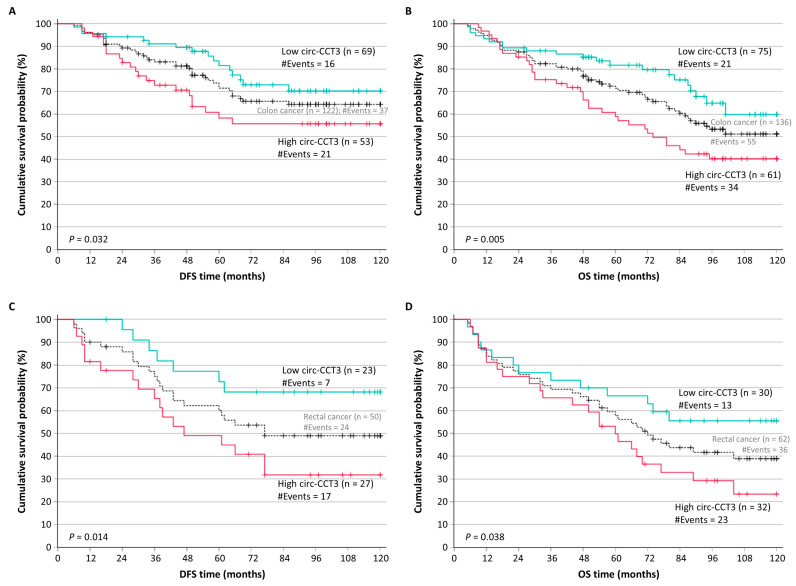
Kaplan–Meier survival curves for the disease-free survival (DFS) and overall survival (OS) of colorectal adenocarcinoma patients, stratified based on their tumor location. Patients with colon adenocarcinoma overexpressing circ-CCT3 had poorer DFS (**A**) and OS (**B**) than those with colon adenocarcinoma with lower circ-CCT3 levels. Similarly, patients with rectal adenocarcinoma overexpressing circ-CCT3 had shorter DFS (**C**) and OS (**D**) time intervals than those with rectal adenocarcinoma showing lower expression of circ-CCT3. The *p* values were estimated by the Mantel–Cox (log-rank) test.

**Table 1 biomedicines-13-02432-t001:** The primers used in the pre-amplification and real-time qPCR assays for the selective amplification of circ-CCT3 and *GAPDH* mRNA.

Assay	Target	Sequence (5′ → 3′)	Ta (°C)
Pre-amplification	circ-CCT3	TCTGCTCGTCTTCCAACATC	58
		GCTCAGGATTATCTGGAAGACC	
	*GAPDH* mRNA	CCACATCGCTCAGACACCAT	60
		TGACAAGCTTCCCGTTCTCA	
Real-time qPCR	circ-CCT3	TACCCAGTCTTCCATCAACTGG	60
		ACACAGGTGCCATCGGAAAC	
	*GAPDH* mRNA	ATGGGGAAGGTGAAGGTCG	60
		GGGTCATTGATGGCAACAATATC	

Abbreviation: T_a_, annealing temperature.

**Table 2 biomedicines-13-02432-t002:** Clinicopathological features of patients included in the current study.

Variable	Number of Patients (%)
**Gender**	
Male	113 (52.3%)
Female	103 (47.7%)
**Histological grade**	
I	19 (8.8%)
II	166 (76.8%)
III	31 (14.4%)
**T (tumor invasion)**	
T1	6 (2.8%)
T2	25 (11.6%)
T3	134 (62.0%)
T4	51 (23.6%)
**N (nodal status)**	
N0	120 (55.6%)
N1	56 (25.9%)
N2	40 (18.5%)
**M (distant metastasis)**	
M0	188 (87.0%)
M1	28 (13.0%)
**TNM stage**	
I	28 (13.0%)
II	86 (39.7%)
III	74 (34.3%)
IV	28 (13.0%)
**Radiotherapy**	
No	169 (83.7%)
Yes	33 (16.3%)
**Chemotherapy**	
No	85 (42.1%)
Yes	117 (57.9%)

Abbreviation: TNM, tumor, node, and metastasis.

**Table 3 biomedicines-13-02432-t003:** circ-CCT3 expression levels in colorectal adenocarcinomas and non-cancerous colorectal tissues.

Variable	Mean ± SEM	Range	Quartiles		
			1st	2nd (Median)	3rd
circ-CCT3 expression (RQU)					
In cancerous tissues (*n* = 216)	5.13 ± 0.123	2.07–7.97	3.46	5.12	6.88
In normal tissues (*n* = 86)	2.25 ± 0.099	0.814–3.76	1.54	2.17	3.07

Abbreviations: RQUs, relative quantification units; SEM, standard error of the mean.

**Table 4 biomedicines-13-02432-t004:** Multivariate Cox regression analysis results examining the independence of circ-CCT3 expression status with regard to colorectal adenocarcinoma patients’ disease-free survival.

	Covariate	HR	95% CI	*p* Value ^1^	BCa 95% CI	Bootstrap *p* Value ^1^
**Univariate analysis**	circ-CCT3 expression (continuous variable)	1.25	1.08–1.43	*0.002*	1.09–1.44	*0.003*
	circ-CCT3 expression status					
	Low	1.00				
	High	2.35	1.40–3.95	*0.001*	1.42–4.13	*0.001*
	Tumor location					
	Colon	1.00				
	Rectum	1.64	0.98–2.74	0.059	0.98–2.71	0.052
	Histological grade					
	I	1.00				
	II	1.73	0.62–4.81	0.29	0.74–9.10	0.24
	III	3.06	0.97–9.61	0.056	1.04–19.73	*0.035*
	TNM stage					
	I	1.00				
	II	4.37	1.03–18.49	*0.045*	1.47–2.65 × 10^4^	*0.029*
	III	9.95	2.39–41.47	*0.002*	3.30–6.85 × 10^4^	*0.001*
	Radiotherapy					
	No	1.00				
	Yes	1.78	0.99–3.19	0.052	0.97–3.10	*0.044*
	Chemotherapy					
	No	1.00				
	Yes	2.32	1.32–4.06	*0.003*	1.31–4.31	*0.004*
**Multivariate analysis**	circ-CCT3 expression status					
	Low	1.00				
	High	1.75	1.03–2.98	*0.039*	1.04–3.27	*0.044*
	Tumor location					
	Colon	1.00				
	Rectum	2.47	1.22–5.00	*0.012*	1.12–5.91	*0.012*
	TNM stage			*0.005*		
	I	1.00				
	II	4.69	1.04–21.19	*0.045*	1.45–3.03 × 10^4^	*0.026*
	III	9.32	1.94–44.67	*0.005*	2.60–7.67 × 10^4^	*0.002*
	Radiotherapy					
	No	1.00				
	Yes	0.55	0.25–1.24	0.15	0.20–1.28	0.19
	Chemotherapy					
	No	1.00				
	Yes	1.30	0.69–2.43	0.42	0.63–2.58	0.44

^1^ Statistically significant *p* values are shown in italics. Abbreviations: BCa, bias-corrected and accelerated; CI, confidence interval; HR, hazard ratio.

**Table 5 biomedicines-13-02432-t005:** Multivariate Cox regression analysis results examining the independence of circ-CCT3 expression status with regard to colorectal adenocarcinoma patients’ overall survival.

	Covariate	HR	95% CI	*p* Value ^1^	BCa 95% CI	Bootstrap *p* Value ^1^
**Univariate analysis**	circ-CCT3 expression (continuous variable)	1.27	1.13–1.43	*<0.001*	1.14–1.44	*0.001*
	circ-CCT3 expression status					
	Low	1.00				
	High	2.12	1.39–3.25	*<0.001*	1.39–3.30	*0.002*
	Tumor location					
	Colon	1.00				
	Rectum	1.53	1.01–2.33	*0.047*	0.98–2.39	0.056
	Histological grade					
	I	1.00				
	II	1.32	0.61–2.88	0.48	0.61–3.61	0.51
	III	2.72	1.14–6.48	*0.024*	1.17–7.56	*0.023*
	TNM stage					
	I	1.00				
	II	2.60	0.91–7.48	0.075	1.03–13.44	0.062
	III	5.76	2.03–16.29	*0.001*	2.30–28.71	*0.005*
	IV	34.85	11.71–103.72	*<0.001*	13.78–203.79	*0.001*
	Radiotherapy					
	No	1.00				
	Yes	1.67	1.02–2.75	*0.042*	0.97–2.84	0.057
	Chemotherapy					
	No	1.00				
	Yes	1.78	1.15–2.77	*0.010*	1.14–2.87	*0.013*
**Multivariate analysis**	circ-CCT3 expression status					
	Low	1.00				
	High	1.26	0.80–1.98	0.33	0.80–2.12	0.35
	Tumor location					
	Colon	1.00				
	Rectum	1.38	0.80–2.38	0.25	0.72–2.74	0.30
	TNM stage			*<0.001*		
	I	1.00				
	II	3.04	1.02–9.04	0.046	1.08–19.36	*0.040*
	III	7.02	2.27–21.74	*<0.001*	2.39–47.43	*0.003*
	IV	40.12	12.15–132.49	*<0.001*	12.67–307.01	*0.001*
	Radiotherapy					
	No	1.00				
	Yes	1.16	0.60–2.24	0.65	0.49–2.52	0.68
	Chemotherapy					
	No	1.00				
	Yes	0.71	0.42–1.18	0.19	0.41–1.27	0.20

^1^ Statistically significant *p* values are shown in italics. Abbreviations: BCa, bias-corrected and accelerated; CI, confidence interval; HR, hazard ratio.

## Data Availability

The data presented in this study are available upon reasonable request from the corresponding authors.

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
