# Peer review of "A Retrospective Study in Colorectal Adenocarcinoma Uncovers the Potential of Circ-CCT3 as a Predictor of Tumor Recurrence"

_biomedicines, 2025, doi:10.3390/biomedicines13102432_

Round 1
Reviewer 1 Report (Previous Reviewer 3)
Comments and Suggestions for Authors
This article explores the diagnostic and prognostic value of circ-CCT3 in colorectal adenocarcinoma. Based on tissue samples from 216 patients, combined with RT-qPCR quantification and survival analysis, it proposes that circ-CCT3 may serve as a potential molecular biomarker for predicting short-term recurrence. The study features a moderate sample size with strong clinical relevance. However, the manuscript exhibits limited overall innovation, with insufficient research design and methodological validation. Statistical interpretation and clinical translational significance require further enhancement.
1.I am concerned that the study is limited by the lack of an external independent validation cohort. Inclusion of data from publicly available resources such as TCGA or GEO, or validation in a multi-center cohort, would greatly strengthen the robustness and generalizability of the findings.
2.I note that the manuscript does not present circ-CCT3 expression stratified across different clinical or molecular subgroups (e.g., tumor sidedness, MSI status, CMS subtypes). Such analyses would provide additional insights into the biological and clinical heterogeneity of colorectal cancer.
3.I am concerned that the methodological validation of circ-CCT3 quantification is incomplete. Key experiments such as RNase R digestion, Sanger sequencing of the back-splice junction, and confirmation of circRNA-specific amplification should be reported to ensure technical reliability.
4.The title currently emphasizes “short-term recurrence predictor,” which may appear overstated. A more precise and balanced phrasing would strengthen credibility.
5.The Discussion section is somewhat lengthy and contains extensive background material. A more concise synthesis, with sharper focus on the novelty of this study, would improve readability.
Author Response
Reviewer’s Comments and Corresponding Responses
- I am concerned that the study is limited by the lack of an external independent validation cohort. Inclusion of data from publicly available resources such as TCGA or GEO, or validation in a multi-center cohort, would greatly strengthen the robustness and generalizability of the findings.
We agree with the Reviewer that validation in a multi-center cohort, would greatly strengthen the robustness and generalizability of the findings. However, since such a cohort is not available in our lab biobank, we have to admit that this is a limitation of our study:
Page 14 (lines 442-444): This is a retrospective, single-center study, based on colorectal tissue samples. Prospective multi-center validation in larger cohorts is prerequisite to confirm clinical utility.
Inclusion of data from publicly available resources such as TCGA or GEO is probably not a way to address this issue in the current study, as such resources do not contain circRNA-seq data. circRNA profiling using next-generation sequencing has a prerequisite: the digestion of linear RNAs included in the total RNA extracts with RNase R, prior to first-strand cDNA synthesis. Publicly available datasets from CRC patients’ samples (along with clinical characteristics) are not included in TCGA, GEO, or other similar public repositories.
- I note that the manuscript does not present circ-CCT3 expression stratified across different clinical or molecular subgroups (e.g., tumor sidedness, MSI status, CMS subtypes). Such analyses would provide additional insights into the biological and clinical heterogeneity of colorectal cancer.
We totally agree with the Reviewer; still, the categorization into CMS was not feasible in our cohort based on the data we had at our disposition, as most samples had been collected more than 10 years ago. This limitation is clearly discussed in the Discussion section, as shown below:
Page 14 (lines 444-447): Another limitation of this research study is the lack of information regarding APC, KRAS, and BRAF mutational status, MSI status, and CIMP, which did not allow categorization according to the consensus molecular subtypes (CMS) widely used in clinical practice, today [45].
Accordingly, we have added the following references:
- Valenzuela, G.; Canepa, J.; Simonetti, C.; Solo de Zaldívar, L.; Marcelain, K.; González-Montero, J. Consensus molecular subtypes of colorectal cancer in clinical practice: A translational approach. World J Clin Oncol 2021, 12, 1000-1008. https://doi.org/10.5306/wjco.v12.i11.1000.
- I am concerned that the methodological validation of circ-CCT3 quantification is incomplete. Key experiments such as RNase R digestion, Sanger sequencing of the back-splice junction, and confirmation of circRNA-specific amplification should be reported to ensure technical reliability.
We thank the Reviewer for this remark. Since the optimization of the qPCR assay, which included ensuring technical reliability, has already been published in a previous study of members of our group, we cannot present such data as new; however, we mention this in detail in the revised version:
Page 4 (lines 168-172): Previous real-time qPCR assay optimization included designing specific divergent primers for circ-CCT3 pre-amplification and qPCR as well as convergent primers for GAPDH amplification, optimization of the conditions and standardization of the molecular quantification assays, and Sanger sequencing of the amplicon spanning circ-CCT3 back-splice junction [23].
The respective reference is the following one:
- Papatsirou, M.; Kontos, C.K.; Ntanasis-Stathopoulos, I.; Malandrakis, P.; Sideris, D.C.; Fotiou, D.; Liacos, C.I.; Gavriatopoulou, M.; Kastritis, E.; Dimopoulos, M.A., et al. Exploring the molecular biomarker utility of circCCT3 in multiple myeloma: A favorable prognostic indicator, particularly for R-ISS II patients. Hemasphere 2024, 8, e34. https://doi.org/10.1002/hem3.34.
Moreover, we added the following sentence in the Materials and Methods section:
Page 5 (lines 189-190): Additionally, melting curves were generated to confirm circRNA-specific amplification in each sample.
We should note that RNase R digestion of linear RNAs was not performed, because a poly(A) RNA (i.e. GAPDH mRNA) was chosen as reference. In fact, there are no circRNAs that are considered as reliable references for relative quantification. However, as we noticed during the optimization of the qPCR assay, the use of divergent primers during both pre-amplification and real-time qPCR amplification ensured the production of specific amplicons by circ-CCT3; the identity of the amplicon in each sample was verified my melting curve analysis.
- The title currently emphasizes “short-term recurrence predictor,” which may appear overstated. A more precise and balanced phrasing would strengthen credibility.
The Reviewer’s suggestion is highly appreciated; we modified the title, as shown below:
Page 1 (new Title): A retrospective study in colorectal adenocarcinoma uncovers the potential of circ-CCT3 as a predictor of tumor recurrence
We truly believe that the new title is more accurate and thank the Reviewer for this remark.
- The Discussion section is somewhat lengthy and contains extensive background material. A more concise synthesis, with sharper focus on the novelty of this study, would improve readability.
According to the Reviewer’s suggestion, we made the Discussion shorter and focused more on the novelty of our study. Therefore:
- We moved a whole paragraph to the Introduction section (lines 70-83 of the revised manuscript).
- We removed part of another paragraph from the Discussion.
- We discussed our findings more thoroughly, as presented below:
Pages 13-14 (lines 390-409): It has recently been suggested that emerging biomarkers such as circRNAs and long non-coding RNAs (lncRNAs) offer opportunities for early CRC detection [35]. Beyond its potential diagnostic utility, higher circ-CCT3 expression levels were associated with higher TNM stage and node involvement of CRC patients, indicating that circ-CCT3 levels increase progressively with disease severity. This finding is in consistence with the modulation of VEGFA and Wnt/β-catenin signaling by circ-CCT3 to enhance CRC metastasis through miRNA sponging, which has recently been documented [24].
Based on findings presented in the current study, it would be tempting to detect and quantify circ-CCT3 levels in extracellular vesicles of CRC patients. Taking it a step further, it would be very interesting to study circ-CCT3 function and role in CRC. In fact, only a few circRNAs found in extracellular vesicles and produced specifically by cancerous or other cancer-associated cells, such as CRC-associated fibroblasts have been studied so far. Interestingly, such circRNAs have very important regulatory roles, also affecting post-transcriptional modifications of mRNAs [36]. Beyond its expression levels, study of N6-methyladenosine (m6A) modifications of circ-CCT3 could prove very interesting, as this dynamic modification system participates in virtually all aspects of circRNA metabolism, including biogenesis, intracellular trafficking, and turnover [37]. By integrating mechanistic insights with preclinical results regarding circ-CCT3 expression in CRC, clinical translation barriers could be addressed.
Page 14 (lines 424-431): Therefore, quantification of circ-CCT3 expression levels could act as a surrogate biomarker in addition to TNM staging, for the determination of a subgroup of patients with a higher probability of disease recurrence. While the unfavorable prognostic value of circ-CCT3 status is in-dependent of important prognosticators with regard to DFS, this is not the case for prediction of OS. The fact that the unfavorable prognostic value of circ-CCT3 status did not prove to be independent in the multivariate Cox regression analysis could be attributed to its significant association with TNM staging, which is a very strong prognosticator of OS.
Pages 14-15 (lines 447-451): Lack of information about tumor budding, a well-established independent prognostic factor in CRC [46], is another limitation of our study. This is due to the fact that most patients providing samples for this study were diagnosed and had their colorectal tumor removed before the establishment of this strong prognosticator.
Pages 15 (lines 453-459): Nevertheless, it should be considered that cancer represents an ecological process, as well. Addressing clinical challenges such as recurrence and treatment resistance may re-quire perspectives that extend beyond genetic and molecular mechanisms alone. For instance, it is common knowledge that polymorphic microbiomes play important roles in CRC occurrence and development [47]. Additionally, information on the association between socioeconomic status and cancer is useful for policy-based CRC control [48].
Accordingly, we have added the following references:
- Zamanian, M.Y.; Darmadi, D.; Darabi, R.; Fanoukh Aboqader Al-Aouadi, R.; Sadeghi Ivraghi, M.; Küpeli Akkol, E. Biomarkers for colorectal cancer detection: An insight into colorectal cancer and FDA-approved biomarkers. Bioimpacts 2025, 15, 31211. https://doi.org/10.34172/bi.31211.
- Tan, J.N.; Yu, J.H.; Hou, D.; Xie, Y.Q.; Lai, D.M.; Zheng, F.; Yang, B.; Zeng, J.T.; Chen, Y.; Lu, S.H., et al. Extracellular Vesicle-Packaged circTAX1BP1 from Cancer-Associated Fibroblasts Regulates RNA m6A Modification through Lactylation of VIRMA in Colorectal Cancer Cells. Adv Sci (Weinh) 2025, 10.1002/advs.202514008, e14008. https://doi.org/10.1002/advs.202514008.
- Xu, J.; Liu, J.; Huang, F.; Chao, Z.; Chen, J.; Xu, A.; Zhang, H.; Zhu, W. Research progress of circular RNA and N6-methyladenosine modification in colorectal cancer. Gene 2025, 967, 149684. https://doi.org/10.1016/j.gene.2025.149684.
- Lugli, A.; Kirsch, R.; Ajioka, Y.; Bosman, F.; Cathomas, G.; Dawson, H.; El Zimaity, H.; Fléjou, J.F.; Hansen, T.P.; Hartmann, A., et al. Recommendations for reporting tumor budding in colorectal cancer based on the International Tumor Budding Consensus Conference (ITBCC) 2016. Mod Pathol 2017, 30, 1299-1311. https://doi.org/10.1038/modpathol.2017.46.
- Zhou, Z.; Yang, M.; Fang, H.; Zhang, B.; Ma, Y.; Li, Y.; Liu, Y.; Cheng, Z.; Zhao, Y.; Si, Z., et al. Tailoring a Functional Synthetic Microbial Community Alleviates Fusobacterium nucleatum-infected Colorectal Cancer via Ecological Control. Adv Sci (Weinh) 2025, 12, e14232. https://doi.org/10.1002/advs.202414232.
- Kanda, S.; Watanabe, K.; Nakamura, S.; Narimatsu, H. Association between socioeconomic background and cancer: An ecological study using cancer registry and various community socioeconomic status indicators in Kanagawa, Japan. PLoS One 2025, 20, e0326895. https://doi.org/10.1371/journal.pone.0326895.
The Authors wish to thank the Reviewers for their constructive comments that led to the improvement of the current manuscript.

Reviewer 2 Report (New Reviewer)
Comments and Suggestions for Authors
Some main points should be concerned as below,
1)The title "circ-CCT3 as a predictor of short-term disease recurrence" ”and line 33-35 “circ-CCT3 overexpression... , independently predicting
short-term disease recurrence”,can the authors show clearly why circ-CCT3 act as a predictor of short-term disease recurrence? From the results of this study, it seems that there is no basis for this.
2)Line 42-43“placing CRC as the second leading cause of cancer-related mortality and highlighting its significant lethality”, it is suggested that the references cited be updated.
3)Tumor budding in colorectal cancer has been widely recognized and incorporated into international consensus guidelines, such as those from the International Tumor Budding Consensus Conference, and is well-established as a strong prognostic factor associated with lymph node metastasis, disease recurrence, and poor survival. Given its clinical significance, the absence of tumor budding in the assessment of clinicopathological features of colorectal adenocarcinoma—and its exclusion from subsequent analyses, including Table 4—raises concerns about the comprehensiveness of the pathological evaluation. The authors are encouraged to clarify whether tumor budding was assessed and, if so, why it was not included in the reported data.
4)The statement in lines 435–436 suggests that circ-CCT3 expression levels may serve as both a diagnostic and prognostic biomarker in colorectal cancer (CRC). However, given that circ-CCT3 is also expressed in normal tissues, its specificity as a diagnostic biomarker remains questionable.
5)The authors are encouraged to read a recent publication and consider incorporating relevant discussions into the manuscript.It is increasingly recognized that cancer represents an ecological process. Addressing clinical challenges such as recurrence and treatment resistance may require perspectives that extend beyond genetic and molecular mechanisms alone.
Author Response
Reviewer’s Comments and Corresponding Responses
- The title "circ-CCT3 as a predictor of short-term disease recurrence" and line 33-35 “circ-CCT3 overexpression... , independently predicting short-term disease recurrence”, can the authors show clearly why circ-CCT3 act as a predictor of short-term disease recurrence? From the results of this study, it seems that there is no basis for this.
The Reviewer’s suggestion is highly appreciated; we modified the title, as shown below:
Page 1 (new Title): A retrospective study in colorectal adenocarcinoma uncovers the potential of circ-CCT3 as a predictor of tumor recurrence
We truly believe that the new title is more accurate and thank the Reviewer for this remark. We also modified the end of the Abstract, as follows:
Page 1 (lines 31-33): In summary, our study provides evidence that circ-CCT3 overexpression constitutes a promising molecular biomarker of poor prognosis in colorectal adenocarcinoma, independently predicting tumor recurrence.
- Line 42-43“placing CRC as the second leading cause of cancer-related mortality and highlighting its significant lethality”, it is suggested that the references cited be updated.
We complied with the Reviewer’s suggestion by removing the previous reference and add three new, suitable references:
- Wang, S.; Zheng, R.; Li, J.; Zeng, H.; Li, L.; Chen, R.; Sun, K.; Han, B.; Bray, F.; Wei, W., et al. Global, regional, and national lifetime risks of developing and dying from gastrointestinal cancers in 185 countries: a population-based systematic analysis of GLOBOCAN. Lancet Gastroenterol Hepatol 2024, 9, 229-237. https://doi.org/10.1016/s2468-1253(23)00366-7.
- Morgan, E.; Arnold, M.; Gini, A.; Lorenzoni, V.; Cabasag, C.J.; Laversanne, M.; Vignat, J.; Ferlay, J.; Murphy, N.; Bray, F. Global burden of colorectal cancer in 2020 and 2040: incidence and mortality estimates from GLOBOCAN. Gut 2023, 72, 338-344. https://doi.org/10.1136/gutjnl-2022-327736.
- Bray, F.; Laversanne, M.; Sung, H.; Ferlay, J.; Siegel, R.L.; Soerjomataram, I.; Jemal, A. Global cancer statistics 2022: GLOBOCAN estimates of incidence and mortality worldwide for 36 cancers in 185 countries. CA Cancer J Clin 2024, 74, 229-263. https://doi.org/10.3322/caac.21834.
- Tumor budding in colorectal cancer has been widely recognized and incorporated into international consensus guidelines, such as those from the International Tumor Budding Consensus Conference, and is well-established as a strong prognostic factor associated with lymph node metastasis, disease recurrence, and poor survival. Given its clinical significance, the absence of tumor budding in the assessment of clinicopathological features of colorectal adenocarcinoma—and its exclusion from subsequent analyses, including Table 4—raises concerns about the comprehensiveness of the pathological evaluation. The authors are encouraged to clarify whether tumor budding was assessed and, if so, why it was not included in the reported data.
Tumor budding was not evaluated in the majority of CRC patients included in this study, as these samples have been collected before 2017. With respect to the Reviewer’s comment, we clarified this limitation in the revised manuscript:
Pages 14-15 (lines 447-451): Lack of information about tumor budding, a well-established independent prognostic factor in CRC [46], is another limitation of our study. This is due to the fact that most patients providing samples for this study were diagnosed and had their colorectal tu-mor removed before the establishment of this strong prognosticator.
Accordingly, we have added the following reference:
- Lugli, A.; Kirsch, R.; Ajioka, Y.; Bosman, F.; Cathomas, G.; Dawson, H.; El Zimaity, H.; Fléjou, J.F.; Hansen, T.P.; Hartmann, A., et al. Recommendations for reporting tumor budding in colorectal cancer based on the International Tumor Budding Consensus Conference (ITBCC) 2016. Mod Pathol 2017, 30, 1299-1311. https://doi.org/10.1038/modpathol.2017.46.
- The statement in lines 435–436 suggests that circ-CCT3 expression levels may serve as both a diagnostic and prognostic biomarker in colorectal cancer (CRC). However, given that circ-CCT3 is also expressed in normal tissues, its specificity as a diagnostic biomarker remains questionable.
According to the Reviewer’s comment, we removed the word “diagnostic” from this concluding sentence; the current sentence is shown below:
Page 15 (lines 461-462): Our study provides robust evidence for the clinical potential of circ-CCT3 expression as a prognostic biomarker in CRC.
- The authors are encouraged to read a recent publication and consider incorporating relevant discussions into the manuscript. It is increasingly recognized that cancer represents an ecological process. Addressing clinical challenges such as recurrence and treatment resistance may require perspectives that extend beyond genetic and molecular mechanisms alone.
We thank utmost the Reviewer for pointing this necessity out. We elaborated on this aspect in the revised version of the manuscript:
Page 15 (lines 453-459): Nevertheless, it should be considered that cancer represents an ecological process, as well. Addressing clinical challenges such as recurrence and treatment resistance may require perspectives that extend beyond genetic and molecular mechanisms alone. For instance, it is common knowledge that polymorphic microbiomes play important roles in CRC occurrence and development [47]. Additionally, information on the association between socioeconomic status and cancer is useful for policy-based CRC control [48].
Accordingly, we have added the following references:
- Zhou, Z.; Yang, M.; Fang, H.; Zhang, B.; Ma, Y.; Li, Y.; Liu, Y.; Cheng, Z.; Zhao, Y.; Si, Z., et al. Tailoring a Functional Synthetic Microbial Community Alleviates Fusobacterium nucleatum-infected Colorectal Cancer via Ecological Control. Adv Sci (Weinh) 2025, 12, e14232. https://doi.org/10.1002/advs.202414232.
- Kanda, S.; Watanabe, K.; Nakamura, S.; Narimatsu, H. Association between socioeconomic background and cancer: An ecological study using cancer registry and various community socioeconomic status indicators in Kanagawa, Japan. PLoS One 2025, 20, e0326895. https://doi.org/10.1371/journal.pone.0326895.
The Authors wish to thank the Reviewers for their constructive comments that led to the improvement of the current manuscript.

Reviewer 3 Report (New Reviewer)
Comments and Suggestions for Authors
Dear authors,
You have done an interesting and innovative work in a very important and evolving field. Nevertheless, there are some points that need further explanations.
- In the introduction, in lines 46 – 47 is stated that: “The primary and potential curative treatment of CRC is surgical resection, followed by adjuvant chemo- and/or radiotherapy”. This is true for colon cancer, because for T3 – T4 rectal cancers neoadjuvant therapy with radiation- and chemotherapy are usually indicated.
- Also, in the introduction the authors assert: “Of note, their circular conformation confers intrinsic ribonuclease resistance, establishing circRNAs as promising clinical biomarkers for non-invasive diagnosis, survival estimation and treatment-response monitoring.” I am wondering which relation can exist between the physico-chemical properties of circRNAs and its possible use as a diagnostic tool or as a biomarker for the determination of life’s expectancy in CRC-patients.
- The previous use of chemo- and/or radiation therapy as an exclusion factor is fully correct in the design of this study. But this could be a biased selection factor for study of the evolution of rectal cancers, because only by T1 or T2 rectal cancers can be treated without neoadjuvant therapy. This implies that in the cohort of rectal cancer patients, only those with smaller tumours were included.
- As the high expression of circRNAs is associated with higher TNM stages, the authors should emphasize which diagnostic or prognostic information can its determination add in the evaluation of CRC patients.
Author Response
Reviewer’s Comments and Corresponding Responses
- In the introduction, in lines 46 – 47 is stated that: “The primary and potential curative treatment of CRC is surgical resection, followed by adjuvant chemo- and/or radiotherapy”. This is true for colon cancer, because for T3 – T4 rectal cancers neoadjuvant therapy with radiation- and chemotherapy are usually indicated.
Prompted by the Reviewer’s comment, we modified this sentence:
Page 2 (lines 43-44): The primary and potential curative treatment of colon cancer is surgical resection, followed by adjuvant chemo- and/or radiotherapy [5,6].
- Also, in the introduction the authors assert: “Of note, their circular conformation confers intrinsic ribonuclease resistance, establishing circRNAs as promising clinical biomarkers for non-invasive diagnosis, survival estimation and treatment-response monitoring.” I am wondering which relation can exist between the physico-chemical properties of circRNAs and its possible use as a diagnostic tool or as a biomarker for the determination of life’s expectancy in CRC-patients.
We thank the Reviewer for this comment. We added a clarifying sentence in the Introduction:
Page 2 (lines 76-79): Of note, their circular conformation confers intrinsic ribonuclease resistance; the pro-longed half-life of circRNAs in bodily fluids as well as their differential expression in cancer patients renders these molecules promising clinical biomarkers for non-invasive diagnosis, survival estimation, and treatment-response monitoring.
Moreover, we added a couple of sentences in the Discussion section, along with the respective citations:
Page 13 (lines 390-392): It has recently been suggested that emerging biomarkers such as circRNAs and long non-coding RNAs (lncRNAs) offer opportunities for early CRC detection [35].
Page 14 (lines 400-409): In fact, only a few circRNAs found in extracellular vesicles and produced specifically by cancerous or other cancer-associated cells, such as CRC-associated fibroblasts have been studied so far. Interestingly, such circRNAs have very important regulatory roles, also affecting post-transcriptional modifications of mRNAs [36]. Beyond its expression levels, study of N6-methyladenosine (m6A) modifications of circ-CCT3 could prove very interesting, as this dynamic modification system participates in virtually all aspects of circRNA metabolism, including biogenesis, intracellular trafficking, and turnover [37]. By integrating mechanistic insights with preclinical results regarding circ-CCT3 expression in CRC, clinical translation barriers could be addressed.
Accordingly, we have added the following references:
- Zamanian, M.Y.; Darmadi, D.; Darabi, R.; Fanoukh Aboqader Al-Aouadi, R.; Sadeghi Ivraghi, M.; Küpeli Akkol, E. Biomarkers for colorectal cancer detection: An insight into colorectal cancer and FDA-approved biomarkers. Bioimpacts 2025, 15, 31211. https://doi.org/10.34172/bi.31211.
- Tan, J.N.; Yu, J.H.; Hou, D.; Xie, Y.Q.; Lai, D.M.; Zheng, F.; Yang, B.; Zeng, J.T.; Chen, Y.; Lu, S.H., et al. Extracellular Vesicle-Packaged circTAX1BP1 from Cancer-Associated Fibroblasts Regulates RNA m6A Modification through Lactylation of VIRMA in Colorectal Cancer Cells. Adv Sci (Weinh) 2025, 10.1002/advs.202514008, e14008. https://doi.org/10.1002/advs.202514008.
- Xu, J.; Liu, J.; Huang, F.; Chao, Z.; Chen, J.; Xu, A.; Zhang, H.; Zhu, W. Research progress of circular RNA and N6-methyladenosine modification in colorectal cancer. Gene 2025, 967, 149684. https://doi.org/10.1016/j.gene.2025.149684.
- The previous use of chemo- and/or radiation therapy as an exclusion factor is fully correct in the design of this study. But this could be a biased selection factor for study of the evolution of rectal cancers, because only by T1 or T2 rectal cancers can be treated without neoadjuvant therapy. This implies that in the cohort of rectal cancer patients, only those with smaller tumours were included.
In our opinion, the inclusion of cases with rectal cancer having been subjected to chemo- and/or radiation therapy would have produced a strong bias in the study. On the other hand, the stratification of patients according to different parameters (one of them being tumor location) decreases the impact of the potential bias mentioned by the Reviewer.
- As the high expression of circRNAs is associated with higher TNM stages, the authors should emphasize which diagnostic or prognostic information can its determination add in the evaluation of CRC patients.
Following the Reviewer’s suggestion, we elaborated on this issue in the Discussion section of the revised manuscript:
Page 14 (lines 424-431): Therefore, quantification of circ-CCT3 expression levels could act as a surrogate biomarker in addition to TNM staging, for the determination of a subgroup of patients with a higher probability of disease recurrence. While the unfavorable prognostic value of circ-CCT3 status is independent of important prognosticators with regard to DFS, this is not the case for prediction of OS. The fact that the unfavorable prognostic value of circ-CCT3 status did not prove to be independent in the multivariate Cox regression analysis could be attributed to its significant association with TNM staging, which is a very strong prognosticator of OS.
The Authors wish to thank the Reviewers for their constructive comments that led to the improvement of the current manuscript.

Round 2
Reviewer 1 Report (Previous Reviewer 3)
Comments and Suggestions for Authors
The authors have meticulously revised the manuscript, demonstrating thorough attention to detail in addressing prior feedback. The current version exhibits a notably high level of scientific rigor, with well-structured content and coherent arguments that enhance readability. Having reviewed the revisions, I am pleased to recommend acceptance of this article.
Reviewer 2 Report (New Reviewer)
Comments and Suggestions for Authors
No other Q
This manuscript is a resubmission of an earlier submission. The following is a list of the peer review reports and author responses from that submission.
Round 1
Reviewer 1 Report
Comments and Suggestions for Authors
Dear Author,
I have read the resubmitted manuscript.
1) Inclusion and exclusion criteria, randomization process.
When the manuscript was submitted for the first time, the authors did not provide any data regarding the randomization process, but this process was reported and explained only after the reviewer's request. This represents a major flaw regarding the reliability of the author's work.
Moreover, the authors did not provide any data regarding the whole selection process (total number of patients screened, number of patients enrolled), nor did they provide a flowchart diagram to explain the randomization process (according to STROBE Guidelines). This enhances the low reliability of the author's selection process and represents another flaw that strongly affects the reliability of the author's work.
"In this way, we tried to address every sentence of the Reviewer’s comment; however, we don’t believe that a flowchart diagram would add value to the manuscript. This would rather repeat information currently provided in detail in the revised text and would be most likely redundant. " I respect the authors' opinion, but this statement is contrary to the current worldwide accepted Guidelines for the publication of studies (e.g., STROBE) and the current literature.
Exclusion criteria included receiving neoadjuvant therapy prior to surgery, missing clinicopathological data, and the existence of known comorbidities at the time of surgery. Inclusion criteria included the diagnosis of primary colorectal adenocarcinoma and sufficient mass of cancerous colorectal tissue available (i.e., 10-50 mg of tissue).
The statement regarding the inclusion and exclusion criteria is ambiguous (e.g., what do you mean by the existence of known comorbidities? Explain which diseases were considered as "comorbidities" and which were not). What about any other possible confounders?
2) Confounders
Thus, it is now evident that there were no patients with comorbidities in the study cohort. We strongly agree with the Reviewer that including such patients and not assessing, at the same time, this parameter would represent a major source of bias that would strongly affect the scientific value of our findings.
Ok, but what about age, BMI, sex, tumor stage?
Did you check for homogeneity regarding baseline characteristics between the two groups?
3) Patients lost to follow-up
I agree that those patients "provided both cancerous and adjacent non-cancerous colorectal tissue specimens," but if your goal is to assess survival, what is the purpose of including patients who were lost very early during follow-up?
4) "A statement regarding the study endpoints and particularly which endpoints and outcome variables were evaluated regarding CCT3 expression is missing." The definition of a primary endpoint is missing; the definition of the secondary endpoints is missing.
5) A statement explaining which groups were compared is missing.
The method section should report something like: we compare group A (patients that have this characteristic) versus group B (patients that had this characteristic); then a subgroup analysis was performed (for this and this reason....) to assess something comparing these patients and these patients.
In conclusion, the after-submission definition of the selection process as randomized represents a strong "red flag" that affects the reliability of the author's work. The lack of a proper methodology (e.g., STROBE Guidelines) strongly affects the scientific value and the methodology of the manuscript. The study design, and both the definition of inclusion/exclusion criteria and the endpoints of the authors' work, were not properly reported. Moreover, it is not clear which groups were compared and if those groups are comparable (according to their baseline characteristics), thus introducing possible confounders and biases.
Reviewer 2 Report
Comments and Suggestions for Authors
Dear authors,
I thank you for the accepted corrections that were suggested to you, and I have no objections to the version of the paper that you have now sent.
Best regards
Reviewer 3 Report
Comments and Suggestions for Authors
This manuscript are promising and potentially impactful, several points require clarification and additional analysis to strengthen the manuscript before it can be considered for publication.
1.Figure 3: The caption should explicitly state the statistical test used (presumably Jonckheere-Terpstra).
2.The discussion should more deeply contextualize the finding that circ-CCT3 is an independent prognostic factor for DFS but not for OS. This is a common occurrence, often because Overall Survival can be influenced by many post-recurrence factors and subsequent lines of therapy. This point should be elaborated upon.
3.The median follow-up time for the cohort is a critical piece of information for survival studies.
4.Title 2.2 There are some considerations regarding total RNA extraction and first-strand cDNA synthesis.
5.Figure 5C, D: The term “Recall cancer” in the figure should be “Rectal cancer.”
6.Please carefully verify the accuracy and consistency of the content.